# Protein language model identifies disordered, conserved motifs implicated in phase separation

Yumeng Zhang[1], Jared Zheng[2], Bin Zhang[1]*

[1]Department of Chemistry, Massachusetts Institute of Technology, Cambridge, United States; [2]Department of Electrical Engineering and Computer Science, Massachusetts Institute of Technology, Cambridge, United States

## eLife Assessment

This **valuable** study presents an analysis of evolutionary conservation in intrinsically disordered regions, identified as key drivers of phase separation, leveraging a protein language model. The strength of evidence presented is **convincing** overall, though the theoretical grounding could benefit from further development.

*For correspondence:
binz@mit.edu

**Abstract** Intrinsically disordered regions (IDRs) play a critical role in phase separation and are essential for the formation of membraneless organelles (MLOs). Mutations within IDRs can disrupt their multivalent interaction networks, altering phase behavior and contributing to various diseases. Therefore, examining the evolutionary constraints of IDRs provides valuable insights into the relationship between protein sequences and phase separation. In this study, we utilized the ESM2 protein language model to map the residue-level mutational tolerance landscapes of IDRs. Our findings reveal that IDRs, particularly those actively participating in phase separation, contain conserved amino acids. This conservation is evident through mutational constraints predicted by ESM2 and supported by direct analyses of multiple sequence alignments. These conserved, disordered amino acids include residues traditionally identified as 'stickers' as well as 'spacers' and frequently form continuous sequence motifs. The strong conservation, combined with their potential role in phase separation, suggests that these motifs may act as functional units under evolutionary selection to support stable MLO formation. Our findings underscore the insights into phase separation's molecular grammar made possible through evolutionary analysis enabled by protein language models.

## Introduction

Membraneless organelles (MLOs), such as nucleoli, stress granules, and P-bodies, are distributed throughout diverse cellular environments and play a vital role in forming specialized biochemical compartments that drive essential cellular functions (*Hirose et al., 2023*; *Banani et al., 2017*; *Choi et al., 2020*; *Pappu et al., 2023*; *Ginell and Holehouse, 2023*; *Latham and Zhang, 2020*; *Latham and Zhang, 2022a*; *Latham and Zhang, 2022c*; *Latham and Zhang, 2022b*; *Latham et al., 2024*; *Liu et al., 2025*; *Lao and Zhang, 2024*; *Jiang et al., 2015*). These biomolecular condensates typically assemble through phase separation, dynamically recruiting reactants and releasing products to improve the efficiency and specificity of cellular processes (*Hyman et al., 2014*; *Feric et al., 2016*; *Pappu et al., 2023*). Intrinsically disordered regions (IDRs), which lack well-defined tertiary structures yet exhibit unique structural disorder, frequently act as scaffolds within MLOs (*Borcherds et al., 2021*; *Hyman et al., 2014*; *Mittag and Pappu, 2022*; *Dignon et al., 2020*;

*Turoverov et al., 2019*). IDRs facilitate multivalent interactions, including π–π stacking, cation–π, and electrostatic interactions, that promote phase separation. Mutations in IDRs can disrupt phase behavior, potentially leading to MLO dysfunction and contributing to diseases such as neurodegenerative disorders and cancer (*Alberti and Dormann, 2019*; *Zbinden et al., 2020*; *Rai et al., 2021*; *Das et al., 2020*).

Substantial research has focused on linking protein sequences to the phase behaviors of condensates (*Brangwynne et al., 2015*; *Pappu et al., 2023*; *Zeng and Pappu, 2023*; *Schmit et al., 2020*; *Mittag and Pappu, 2022*; *Dzuricky et al., 2018*; *Flory, 1942*; *Huggins, 1942*; *Dignon et al., 2018*; *Dignon et al., 2019*; *Latham and Zhang, 2020*; *Joseph et al., 2021*; *Regy et al., 2021*; *Latham and Zhang, 2021*; *Tesei et al., 2021*; *Benayad et al., 2021*; *Latham and Zhang, 2022a*; *Latham and Zhang, 2022b*; *Farag et al., 2023*; *Lotthammer et al., 2024*; *Bülow et al., 2024*; *Zhang et al., 2024*; *Kapoor et al., 2024*; *Liu et al., 2023*). These studies support the 'stickers-and-spacers' framework (*Choi et al., 2020*; *Pappu et al., 2023*; *Wang et al., 2018*; *Prusty et al., 2018*; *Choi et al., 2019*; *Zhang et al., 2021*), in which specific residues, termed 'stickers', drive strong, specific interactions, while 'spacer' regions act as flexible linkers with minimal nonspecific interactions (*Harmon et al., 2025*; *Bremer et al., 2022*; *Farag et al., 2022*; *Farag et al., 2023*; *Ranganathan and Shakhnovich, 2020*). *Sood and Zhang, 2024* further introduced an evolutionary dimension, proposing that IDRs adapt this framework to balance effective phase separation with compositional specificity. For instance, membraneless organelles (MLOs) form through specific interactions among stickers, while minimizing nonspecific interactions among spacers to enable condensate formation with defined structural and compositional properties. This evolutionary pressure supports the enrichment of low-complexity domains in IDRs, reducing nonspecific interactions yet preserving conserved stickers critical for condensate specificity and stability. This evolutionary perspective, while promising, has not been extensively validated.

Evolutionary analysis of IDRs is challenging due to difficulties in sequence alignment (*Riley et al., 2023*; *Oldfield and Dunker, 2014*; *Brown et al., 2002*; *Huang and Sarai, 2012*; *Nunez-Castilla and Siltberg-Liberles, 2020*; *Thompson et al., 2011*; *Lange et al., 2016*), though several studies have attempted alignment of disordered proteins with promising results (*Dasmeh et al., 2022*; *Lu et al., 2022*; *Ho and Huang, 2022*).

Recent advances in protein language models (*Ofer et al., 2021*; *Rives et al., 2021*; *Elnaggar et al., 2021*; *Strodthoff et al., 2020*; *Alley et al., 2019*; *Brandes et al., 2022*) provide alternative approaches for sequence analysis and information decoding. Trained on large protein datasets, such as UniProt (*Boutet et al., 2016*), these models leverage neural network architectures like the Transformer (*Vaswani et al., 2017*) to capture correlations among amino acids within a sequence. These correlations enable the models to identify constraints, enforced by the whole sequence, that influence the chemical identity of amino acids at specific positions. Since these constraints often reflect functional or structural requirements, the mutational preferences predicted by these models can be interpreted as proxies for evolutionary constraint or mutational tolerance (*Meier et al., 2024*; *Cagiada et al., 2024*; *Ouyang-Zhang et al., 2024*; *Lin et al., 2024*; *Saadat and Fellay, 2025*; *Chu et al., 2024*; *Marquet et al., 2024*; *Gong et al., 2023*). Unlike traditional methods, these predictions do not rely on sequence homology alignments, making them particularly attractive for analyzing disordered protein sequences where alignment is unreliable.

While protein language models have been widely applied to structured proteins (*Brandes et al., 2023*; *Lin et al., 2023*; *Zeng et al., 2024*; *Gong et al., 2023*; *Lin et al., 2024*; *Saadat and Fellay, 2025*; *Chu et al., 2024*), it is important to emphasize that these models themselves are not inherently biased toward folded domains. For example, the Evolutionary Scale Model (ESM2) (*Lin et al., 2023*) is trained as a probabilistic language model on raw protein sequences, without incorporating any structural or functional annotations. Its unsupervised learning paradigm enables ESM2 to capture statistical patterns of residue usage and evolutionary constraints without relying on explicit structural information. Thus, the success of ESM2 in modeling the mutational landscapes of folded proteins (*Brandes et al., 2023*; *Lin et al., 2023*; *Gong et al., 2023*; *Lin et al., 2024*; *Saadat and Fellay, 2025*; *Chu et al., 2024*) reflects the model's ability to learn sequence-level constraints imposed by natural selection—a property that is equally applicable to IDRs if those regions are also under functional selection. Indeed, protein language models are increasingly being used to analyze variant effects in IDRs (*Cagiada et al., 2024*; *Brandes et al., 2023*; *Meier et al., 2024*).

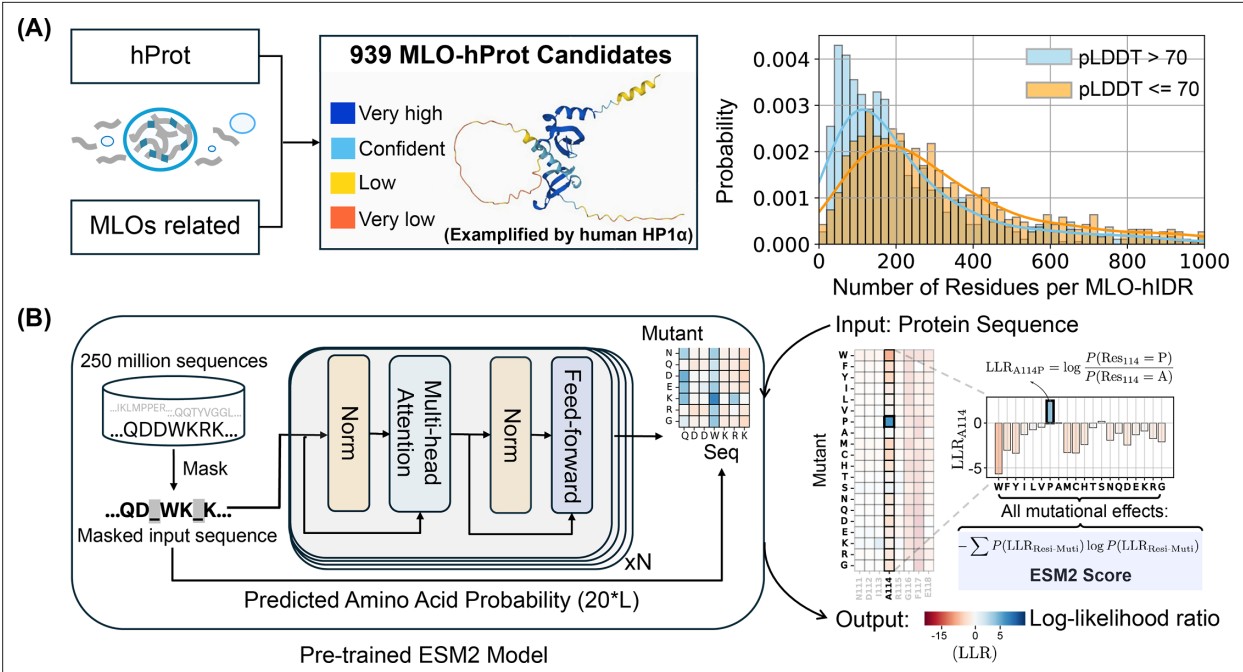

**Figure 1.** Predicting the mutational landscape of intrinsically disordered regions (IDRs) involved in membraneless organelle (MLO) formation (MLO-hProt) using ESM2. (**A**) Schematic representation of the MLO-hProt database construction. The right panel shows the distribution of disordered and folded residues identified in each protein, where structural order is assigned using the AlphaFold2-predicted Local Distance Difference Test (pLDDT) score. (**B**) Workflow of the pretrained ESM2 model (left) (*Lin et al., 2023*) for predicting the mutational landscape of a given protein sequence (right). Upon receiving a protein sequence as input, ESM2 generates a log-likelihood ratio (LLR) for each mutation type at each residue position. Using the 20-element LLR vector, we compute the ESM2 score for each residue (*Equation 1*) to assess mutational tolerance.

In this study, we employ the ESM2 model to investigate the mutational landscape of IDRs involved in MLO formation. Our analysis demonstrates the utility of ESM2 for examining disordered sequences, identifying a notable subset of amino acids that exhibit mutation resistance. These amino acids are evolutionarily conserved, as confirmed through multi-sequence alignment (MSA) analysis. Notably, regions associated with phase separation are significantly enriched in these conserved residues. Importantly, the conserved disordered amino acids include both sticker residues, such as tyrosine (Y), tryptophan (W), and phenylalanine (F), and spacer residues, such as alanine (A), glycine (G), and proline (P). These residues frequently colocalize within continuous sequence stretches, underscoring the functional relevance of entire motifs rather than isolated residues in MLO formation. Our findings provide strong evidence for evolutionary pressures acting on specific IDRs, likely to maintain their roles in scaffolding phase separation mechanisms.

## Results
### Protein language model for quantifying the mutational landscape of MLO proteins

To examine the mutational tolerance of IDRs and their connection to phase separation, we compiled a database of human proteins with disordered regions. From this, we identified a subset of 939 proteins associated with the formation of MLOs, referred to as MLO-hProt. The Methods section provides additional information on the dataset preparation. The dataset contains proteins with varying numbers of disordered residues, ranging from a few dozen to several thousand per protein (*Figure 1A*). These proteins are involved in the assembly of various MLOs, including P-bodies, Cajal bodies, and centrosome granules, and are distributed across both nuclear and cytoplasmic compartments.

We analyzed proteins in the MLO-hProt dataset using the protein language model, ESM2 (*Lin et al., 2023*). As illustrated in *Figure 1B*, ESM2 is a conditional probabilistic model (masked language

model) that predicts the likelihood of specific amino acids appearing at a given position, based on the surrounding sequence context.

The mutational tolerance of a specific amino acid at a given site is defined as follows. ESM2 enables the quantification of the probability, or likelihood, of observing any of the 20 amino acids at site $i$. To assess the preference for a mutant over the wild-type (WT) residue, we calculate the log-likelihood ratio (LLR) between the mutant and WT residues. Consequently, a 20-element vector representing the LLRs for each amino acid can be generated at each site (*Figure 1B*). This vector is then condensed into a single value, referred to as the ESM2 score, which is derived using an information entropy expression for the LLR probabilities of individual amino acids (*Equation 1* in the Methods section).

The ESM2 score provides a measure of the overall mutational tolerance of a given residue. Lower scores indicate higher mutational constraint and reduced flexibility, implying that these residues are more likely essential for protein function, as they exhibit fewer permissible mutational states.

## ESM2 identifies conserved, disordered residues

We next used ESM2 to analyze the mutational tolerance of amino acids in both structured and disordered regions. We carried out ESM2 predictions for all proteins in the MLO-hProt dataset and determined the ESM2 scores of individual amino acids. In addition, to quantify structural disorder, we computed the AlphaFold2-predicted Local Distance Difference Test (pLDDT) scores for each residue. The pLDDT scores have been shown to correlate well with protein flexibility and disorder (*Jumper et al., 2021*), making them a reliable tool for distinguishing structured from unstructured regions. Following previous studies (*Ruff and Pappu, 2021*; *Alderson et al., 2023*), we used a threshold of pLDDT = 70 to differentiate ordered from disordered residues. This threshold reflects amino acid composition preferences for folded versus disordered proteins (see *Figure 2—figure supplement 1*; *Ruff and Pappu, 2021*; *Alderson et al., 2023*).

We first analyzed the relationship between ESM2 and pLDDT scores for human Heterochromatin Protein 1α (HP1α, residues 1–191). HP1α is a crucial chromatin organizer that promotes phase separation and facilitates the compaction of chromatin into transcriptionally inactive regions (*Larson et al., 2017*; *Sanulli et al., 2019*; *Latham and Zhang, 2021*; *Tortora et al., 2023*; *Brennan et al., 2024*). HP1α comprises both structured and disordered segments, as illustrated in *Figure 2A*. Here, residues with pLDDT scores exceeding 70 (indicating ordered regions) are shown in white, while disordered segments (pLDDT ≤70) are highlighted in blue. Similar plots for other proteins are included in *Figure 2—figure supplement 2*. *Figure 2B* displays the AlphaFold2-predicted structure of HP1α, with residues colored according to their pLDDT scores.

Our analysis demonstrated that HP1α's structured domains consistently yield low ESM2 scores, reflecting strong mutational constraints characteristic of folded regions. These constraints are further evident in the local LLR predictions, as shown in *Figure 2B*, where we illustrate the folded region G120–T130. Given the critical role of maintaining the 3D conformation of structured domains, mutations with pronounced deleterious effects are likely to significantly disrupt protein folding. This interpretation aligns with prior studies that have reported a strong correlation between ESM2 LLRs and changes in free energy associated with protein structural stability (*Brandes et al., 2023*; *Lin et al., 2023*).

In contrast, disordered regions, including the N-terminal (residues 1–19), C-terminal (175–191), and hinge domain (70–117), typically exhibit higher ESM2 scores, indicating increased mutational flexibility. Nonetheless, not all disordered regions show similar flexibility. Within the hinge domain, a conserved segment known as the KRK patch (highlighted in orange) (*Smothers and Henikoff, 2001*; *Badugu et al., 2005*; *Larson et al., 2017*; *Strom et al., 2021*) shows low ESM2 scores, despite being disordered. This distinction allows us to classify disordered regions into two types: 'flexible disordered' regions, which show high ESM2 scores and greater mutational tolerance, and 'conserved disordered' regions, which display low ESM2 scores, indicating varying levels of mutational constraint despite a lack of stable folding.

We then examined the distribution of ESM2 scores for all amino acids in the MLO-hProt dataset to evaluate the generality of these patterns. Amino acids in folded regions (pLDDT >70) consistently yield low ESM2 scores, reflecting strong mutational constraints. As shown in *Figure 2C*, the histogram of ESM2 versus pLDDT scores for structured residues reveals a dominant population with low ESM2 values (region a, ESM2 score ≤0.5), consistent with the established understanding that folded domains

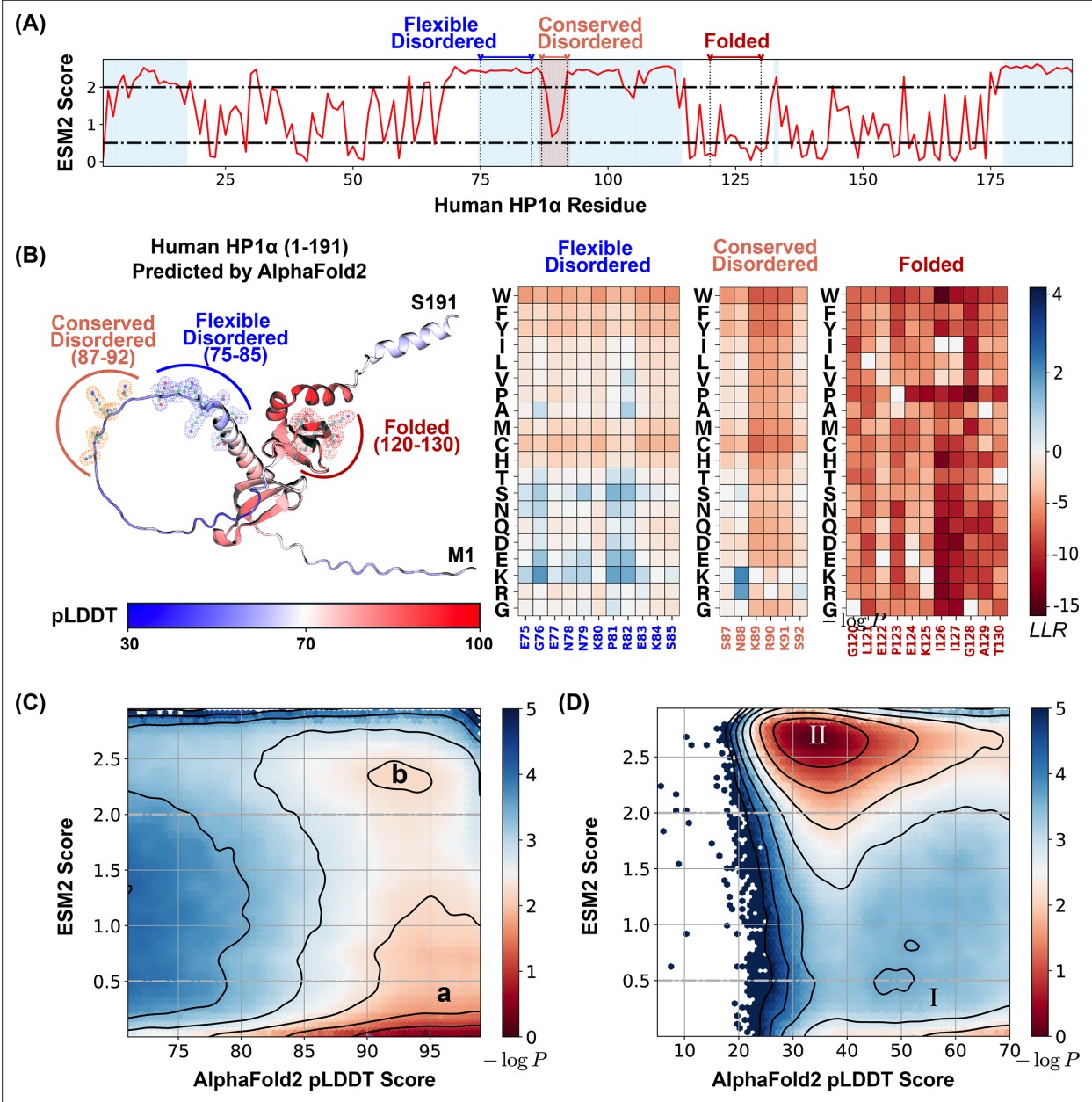

**Figure 2.** ESM2 predicts mutational landscape for structured and disordered residues. (**A**) The ESM2 scores for amino acids in the human HP1α protein (UniProt ID: P45973) are presented, with residues having predicted Local Distance Difference Test (pLDDT) scores below 70 highlighted in blue to signify regions lacking a defined structure. (**B**) A detailed view of the mutational landscape across three regions with varying degrees of structural order. On the left, the AlphaFold2-predicted structure of the human HP1α protein is displayed in cartoon representation, with residues colored according to their pLDDT scores. Three specific regions, representing flexible disordered (residues 75–85), conserved disordered (residues 87–92), and folded (residues 120–130) segments, are highlighted in blue, orange, and red, respectively, using ball-and-stick styles. The panels on the right depict the ESM2 log-likelihood ratio (LLR) predictions for each of these regions. Histograms of pLDDT and ESM2 score distributions for structured (**C**) and disordered (**D**) residues are presented. Contour lines indicate free energy levels computed as $-\log P(\text{pLDDT}, \text{ESM2})$, where $P$ is the probability density of residues based on their pLDDT and ESM2 scores. Contours are spaced at 0.5-unit intervals to distinguish areas of differing density.

The online version of this article includes the following figure supplement(s) for figure 2:

**Figure supplement 1.** Fraction of different amino acids in structured and disordered residues identified from proteins in the MLO-hProt dataset (939).

**Figure supplement 2.** ESM2 and AlphaFold2 predictions for all proteins in the dMLO-hProt dataset.

require structural and functional integrity and are thus more mutation sensitive (*Shakhnovich et al.,* *1996*; *Hamill et al., 2000*; *Ingles-Prieto et al., 2013*).

In contrast, disordered residues (pLDDT ≤70) predominantly show high ESM2 scores (region II, ESM2 score ≥2.0), consistent with the rapid evolution and higher mutational tolerance typical of disordered proteins (*Brown et al., 2011*; *Liu and Huang, 2014*; *Forman-Kay and Mittag, 2013*). However, as shown in *Figure 2D*, a substantial subset of disordered amino acids also exhibits low ESM2 scores (region I). Given that low ESM2 scores generally reflect mutational constraint in folded proteins, the presence of region I among disordered residues suggests that certain disordered amino acids are evolutionarily conserved and likely functionally significant.

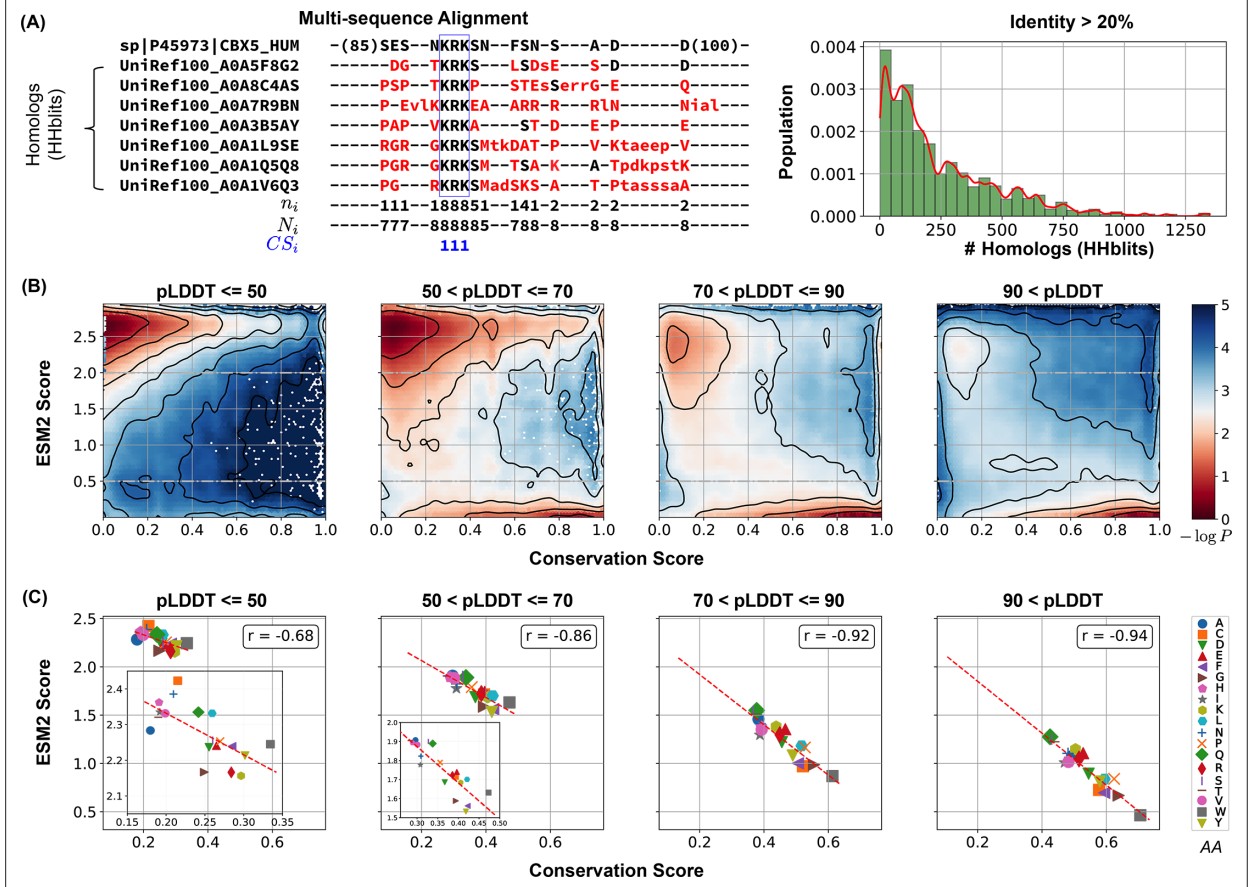

**Figure 3.** Low ESM2 scores correlate strongly with evolutionary conservation. (**A**) Estimating amino acid conservation using multiple sequence alignment. The conservation score calculation is demonstrated for human HP1α protein along with a subset of its homologs found by HHblits. In the aligned sequences, missing residues appear as dashed lines, insertions are shown in lowercase letters, and mismatches are highlighted in red. The three rows below the alignment indicate at position , the number of conserved residues from the reference sequence to the query sequences; $N_i$; the total number of existing residues; and $CS_i$, the conservation score calculated from *Equation 2*, respectively. The right panel illustrates the distribution of homolog counts for each MLO-hProt found by HHblits. (**B**) Histograms showing conservation and ESM2 score distributions for all residues in MLO-hProt, grouped by predicted Local Distance Difference Test (pLDDT) scores from AlphaFold2. The contour lines denote free energy levels, calculated as $-\log P(\text{CS}, \text{ESM2})$, where $P$ is the probability density of residues based on their conservation and ESM2 scores. Contours are spaced at 0.5-unit intervals to highlight regions of distinct density. (**C**) Correlation between mean conservation and ESM2 scores for amino acids classified by structural order levels. Pearson correlation coefficients, $r$, are reported in the legends.

The online version of this article includes the following figure supplement(s) for figure 3:

**Figure supplement 1.** Average ESM2 score for amino acids with different structural order, averaged over residues from proteins in the MLO-hProt dataset (939).

## ESM2 scores correlate with sequence conservation

Our analysis indicates that a substantial proportion of amino acids within disordered regions exhibit low mutational tolerance. To evaluate this hypothesis, we conducted an evolutionary analysis of MLO-hProt proteins, examining the conservation patterns of individual amino acids.

This analysis was based on an MSA of homologs of MLO-hProt proteins. We employed HHblits for homolog detection, a method particularly suited to disordered proteins as it effectively captures distant sequence similarities in highly divergent sequences (*Sharma et al., 2016*; *Jarnot et al., 2022*; *Peng et al., 2023*). The presence of folded domains in these proteins facilitates reliable alignment between references and their query homologs. To exclude sequences that no longer qualify as homologs, we filtered for sequences with at least 20% identity to the reference, resulting in homologous sets ranging from tens to thousands per protein (*Figure 3A*). From these aligned sequences, we calculated the conservation score for each reference amino acid as the ratio of its occurrence in homologs to the total number of sequences (see *Equation 2* in the Methods section).

Our findings reveal a strong correlation between ESM2 scores and conservation scores. In *Figure 3B*, we present the histograms of ESM2 and conservation scores for all amino acids from MLO-hProt proteins. Given that folded domains generally show higher conservation scores than disordered regions, we further classified residues into four groups based on their AlphaFold2 pLDDT scores to assess conservation patterns across varying levels of structural disorder. This stratification allowed us to analyze conservation trends in detail. Across all categories, we observed bimodal distributions, reinforcing the correlation between increasing ESM2 scores and decreasing conservation.

The conservation of amino acids with low ESM2 scores is also apparent in *Figure 3C*. For each of the four structural order groups, we computed the average ESM2 and conservation scores for all 20 amino acid types. Methionine (M) was excluded from the correlation analysis due to its frequent position as the initial residue in sequences, which complicates reliable mutational effect predictions (*Sagawa et al., 2024*; *Meier et al., 2024*). In each group, we consistently observed a strong correlation between average ESM2 and conservation scores.

While ESM2 scores align closely with conservation scores, the relative conservation of specific amino acids varies across structural order groups. In more disordered regions, hydrophilic residues such as glutamine (Q), lysine (K), and arginine (R) exhibit lower ESM2 scores, indicating that mutations in these residues are particularly detrimental. Conversely, hydrophobic residues like valine (V) and isoleucine (I) show higher ESM2 scores, suggesting they experience reduced evolutionary constraints. In more folded domains, hydrophobic residues such as W and F are more conserved (see *Figure 3—figure supplement 1*), consistent with the characteristic conservation patterns of proteins across different disorder levels (*Yang et al., 2023*; *Chavali et al., 2020*). Overall, these findings strongly support our hypothesis that ESM2 scores effectively capture evolutionary conservation, enabling the identification of functionally significant residues through the mutational landscape, independent of structural flexibility.

## Regions driving phase separation are enriched with conserved, disordered residues

The presence of evolutionarily conserved disordered residues raises the question of their functional significance. To explore this, we identified disordered regions of MLO-hProt using a pLDDT score ≤70 and partitioned these regions into two categories: drivers (dMLO-hIDR), which actively drive phase separation, and clients (cMLO-hIDR), which are present in MLOs under certain conditions but do not promote phase separation themselves (*Farahi et al., 2021*). Additionally, IDRs from human proteins not associated with MLOs, termed nMLO-hIDR, were included as a control. To enhance statistical robustness, we extended our dataset by incorporating driver proteins from additional species (*Orti et al., 2024*), resulting in the expanded dMLO-IDR dataset. *Figure 4—figure supplement 1* shows the amino acid composition across these datasets. Beyond the pLDDT-based classification, the majority of residues in these datasets are also predicted to be disordered by various computational tools and supported by experimental evidence (*Figure 4—figure supplements 2 and 3*).

As illustrated in *Figure 4A*, there is a progressive increase in the fraction of conserved disordered residues and a corresponding decline in flexible disordered residues from non-phase-separating proteins (nMLO-hIDR) to clients (cMLO-hIDR) and drivers (dMLO-hIDR and dMLO-IDR) (see also *Figure 4—figure supplement 4*). Driver proteins, particularly those in the expanded dataset, display

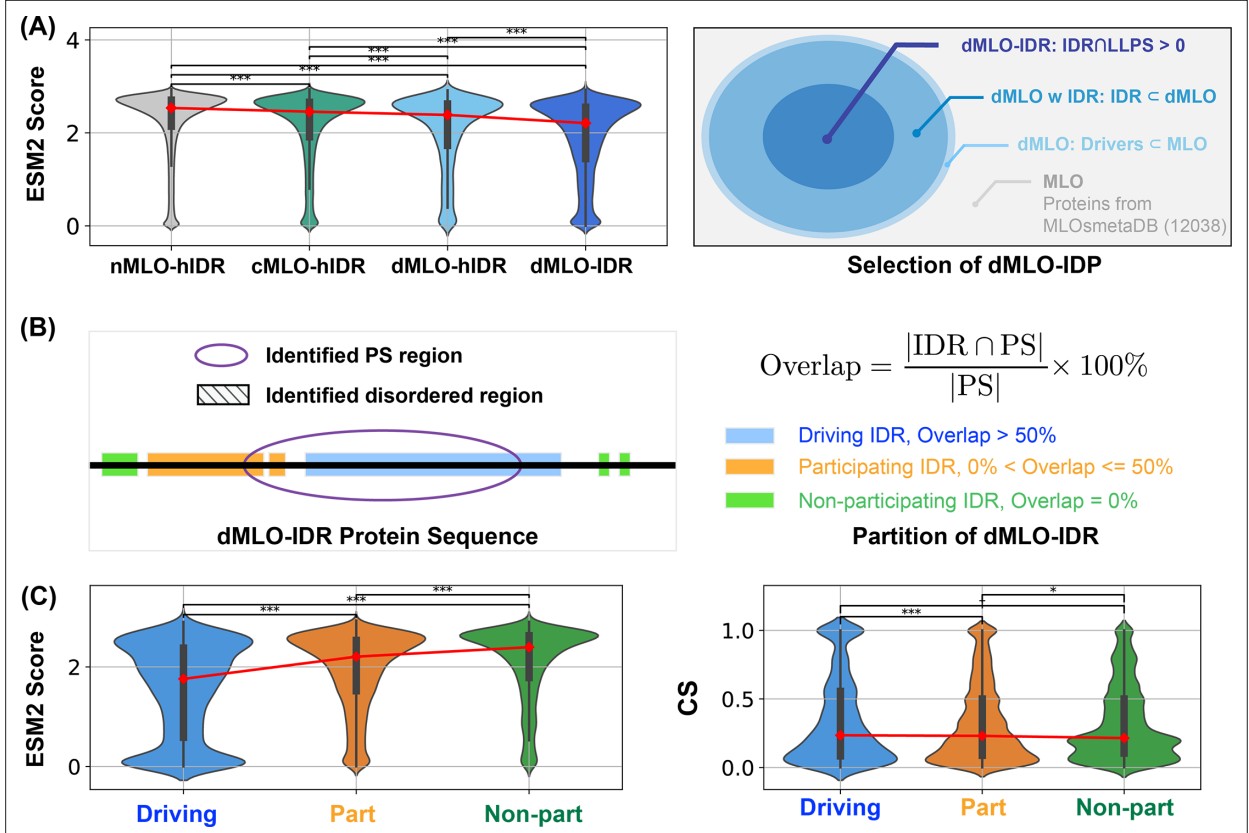

**Figure 4.** Phase separation driving intrinsically disordered regions (IDRs) exhibits more conserved disorder. (**A**) Population of ESM2 score for disordered residues in proteins from nMLO-hIDR, cMLO-hIDR, dMLO-hIDR, and dMLO-IDR datasets. Red dots indicate the mean values of the respective distributions. The selection of proteins in the dMLO-IDR dataset is shown in the right panel. See also methods for details in dataset preparation. (**B**) The classification of three IDR functional groups based on their overlap with the experimentally identified phase separation (PS) segments. (**C**) The distribution of the ESM2 score for residues in three IDR groups, driving (blue), participating (orange), and non-participating (green) shown in the violin plot. The distribution of the conservation score (CS) for residues in three IDR groups shown in the violin plot in the left panel with same coloring scheme as in the right. Pairwise statistical comparisons were conducted using two-sided Mann–Whitney $U$ tests on the ESM2 score distributions (null hypothesis: the two groups have equal medians). p-values indicate the probability of observing the observed rank differences under the null hypothesis. Statistical significance is denoted as follows: ***p < 0.001; **p < 0.01; *p < 0.05; †p < 0.10 (marginal); n.s.: not significant, p ≥ 0.10.

The online version of this article includes the following figure supplement(s) for figure 4:

**Figure supplement 1.** Fraction of different amino acids across proteins from various datasets.

**Figure supplement 2.** Disorder annotation of dMLO-IDR proteins in the MobiDB database (***Di Domenico et al., 2012***).

**Figure supplement 3.** Disorder annotation of human proteins in MobiDB (***Di Domenico et al., 2012***).

**Figure supplement 4.** Violin plots of conservation scores for disordered residues across nMLO-hIDR, cMLO-hIDR, dMLO-hIDR, and dMLO-IDR.

**Figure supplement 5.** Statistics of three intrinsically disordered region (IDR) subgroups in dMLO-IDR.

**Figure supplement 6.** Statistics of conserved amino acids in regions with varying contributions to phase separation.

a notable reduction in flexible residues. These findings imply that disordered regions with a role in phase separation tend to contain functionally significant and evolutionarily conserved regions.

We further examined the sequence location of conserved, disordered residues in driver proteins (dMLO-IDR). For these proteins, experimentally verified segments have been identified, the deletion or mutation of which impairs phase separation (***Mészáros et al., 2020***; ***You et al., 2020***; ***Ning et al., 2020***; ***Li et al., 2020***; ***Orti et al., 2024***; ***Figure 4B***). These segments can include both structured and disordered regions. Herein, if a disordered region constitutes over 50% of the phase-separating segment, we designate it as 'driving', indicating a likely critical contribution to phase separation. If the disordered region represents less than 50%, we classify it as 'participating', with a potentially limited role. Finally, if there is no overlap between the disordered region and the phase-separating segment,

we categorize it as 'non-participating'. The number of segments in the three IDR groups, along with their amino acid compositions, is shown in *Figure 4—figure supplement 5*.

We then analyzed the distribution of ESM2 predictions across these IDR groups. In alignment with *Figure 3A*, we observed a significantly higher proportion of conserved disordered residues within driving IDRs, while few were present in non-participating IDRs. Supporting the ESM2 predictions, conservation analysis based on MSA also indicated that driving IDRs contain a greater concentration of conserved residues (*Figure 4C*). Collectively, these findings demonstrate that ESM2 effectively identifies evolutionarily conserved functional sites, enriched in IDR regions likely involved in driving phase separation.

### Conserved, disordered residues form motifs

Finally, we investigated the chemical identities of conserved residues within driving IDRs to understand their potential role in phase separation. *Figure 5A* displays the average ESM2 LLR predictions for each of the 20 amino acids in the mutational matrix, indicating that mutations to most amino acids are generally unfavorable, as reflected by their low, negative LLR values. This trend is particularly pronounced in driving IDRs compared to nMLO-IDRs or non-participating IDRs (*Figure 5—figure supplement 1*).

We further characterize these conserved residues within driving IDRs. Using hierarchical clustering on two UMAP-derived embeddings from the LLR vectors, we grouped amino acids into five clusters (*Figure 5B*). This approach distinguishes more conserved residues (Groups I–III) from the more flexible residues (Groups IV and V). Notably, W, F, and Y—often referred to as 'stickers' due to their crucial role in phase separation (*Wang et al., 2018*; *Saar et al., 2021*; *Ozawa et al., 2023*; *Rekhi et al., 2024*)— are uniquely grouped within the highly conserved Group I. These findings support the expectation that amino acids essential to phase separation are often evolutionarily conserved, aligning with their central role in functional stability.

Interestingly, residues in Groups II and III, which include traditional 'spacers' (G, A, P, and S), also show high conservation and resistance to most mutation types, particularly hydrophobic mutations (*Figure 5A*). Spacer residues, generally regarded as less critical for interactions driving condensate formation, were unexpectedly conserved, suggesting a broader functional relevance than previously assumed.

We propose that this conservation pattern for spacers is likely not due to isolated residue conservation but may instead reflect the conservation of specific sequence stretches. To examine this hypothesis, we identified ESM2 'motifs' as continuous sequence regions with average ESM2 scores below 0.5. A full list of motifs is available in the (ESM2_motif_with_exp_ref.csv). We observed that conserved amino acids with ESM2 scores below 0.5 are predominantly located within these motifs (see *Figure 5C*, *Figure 5—figure supplement 2A*). For instance, conserved glycine residues have a 97.9% likelihood of being part of an ESM2 motif, with similar probabilities observed for other spacers, such as alanine (99.0%) and proline (93.1%), as well as for sticker residues like Y, W, and F.

These results suggest that IDRs crucial for phase separation frequently contain conserved sequence motifs composed of both sticker and spacer residues. Interestingly, many of these motifs have been experimentally validated as essential for phase separation, with representative motifs for each driving IDR listed in ESM2_motif_with_exp_ref.csv. In these cases, mutations or deletions have been shown to disrupt phase separation. For visualization, a word cloud of these motifs is presented in *Figure 5D*. Altogether, our analysis suggests a tendency for IDRs to uniquely cluster conserved residues into motifs and execute significant biological roles in phase separation.

## Discussion

We have utilized the protein language model ESM2 to investigate the mutational landscape of IDRs. Our analysis reveals a substantial population of mutation-resistant amino acids. MSA confirms their evolutionary conservation. Notably, regions actively involved in phase separation are enriched with these conserved, disordered residues, supporting their potential role in the formation of MLOs. These findings underscore evolutionary constraints on specific IDRs to preserve their functional roles in scaffolding phase separation processes.

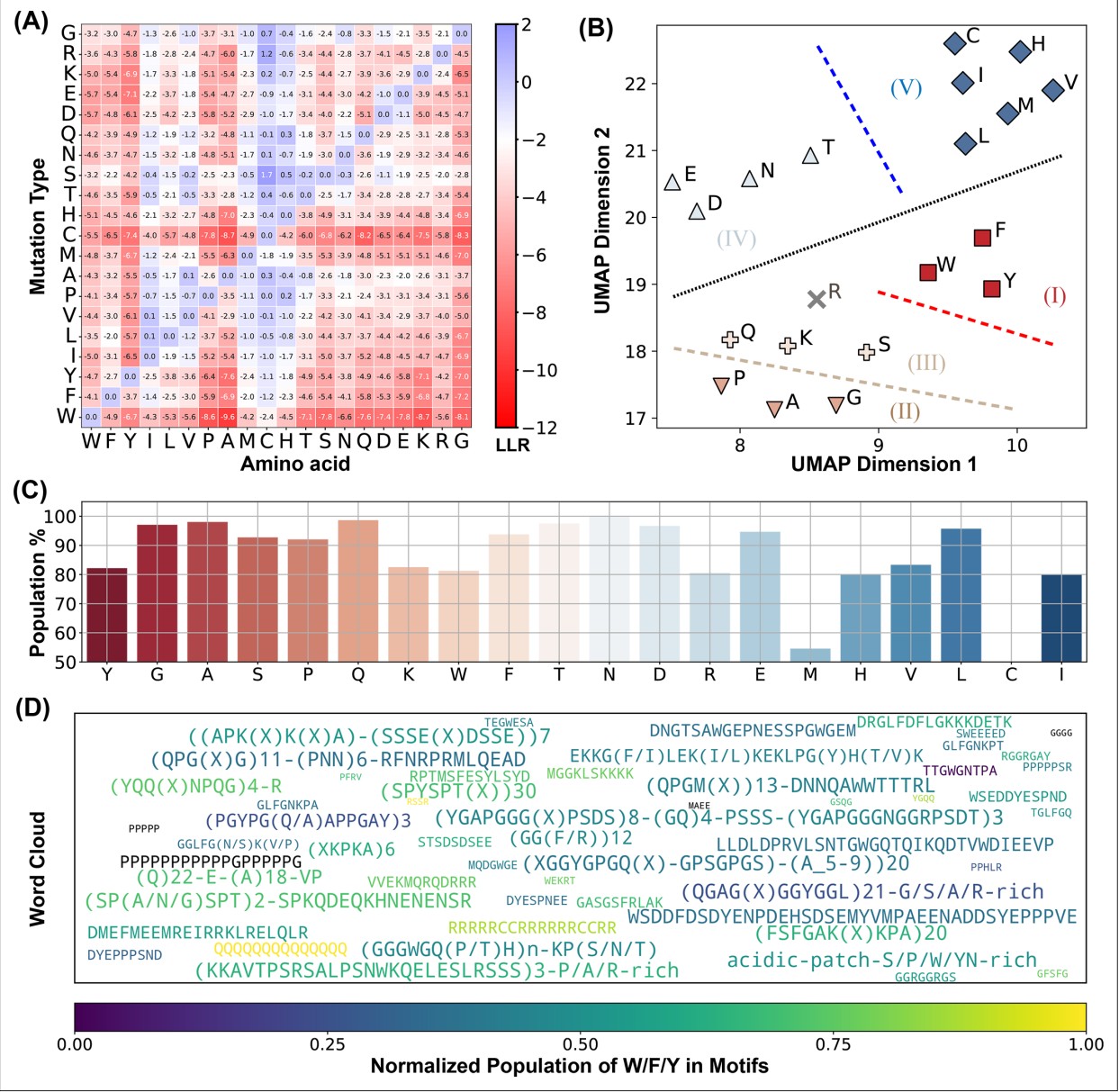

**Figure 5.** ESM2 identifies conserved motifs in driving intrinsically disordered regions (IDRs). (**A**) Mean log-likelihood ratio (LLR) values for the 20 amino acids calculated by averaging across all residues of each amino acid type. (**B**) Clustering of amino acids based on the two UMAP embeddings of their LLR vectors presented in part A. The dashed lines are manually added for a clear visualization of the separation of each group. (**C**) The percent of conserved residues locating in motifs for all amino acids. (**D**) Word cloud of motifs identified by ESM2. The word font size reflects the relative motif length, while the color represents the proportion of 'sticker' residues (Y, F, W, R, K, and Q) within each motif.

The online version of this article includes the following figure supplement(s) for figure 5:

**Figure supplement 1.** Mean log-likelihood ratio (LLR) values across 20 amino acids.

**Figure supplement 2.** Distribution of conserved residues in motifs for driving intrinsically disordered regions (IDRs).

**Figure supplement 3.** ESM2 predictions as indicators of pathogenic mutations.

We emphasize that the results presented in *Figure 4* do not directly demonstrate that conserved residues are preferentially located in regions associated with phase separation. Although these regions are more enriched in conserved amino acids, their total sequence length can be smaller than that of non-phase-separating regions. As a result, the absolute number of conserved residues may still be higher outside phase-separating regions. To quantitatively assess this, we calculated, for each protein in the MLO-hProt dataset, the probability $p$ of finding conserved residues within regions contributing

to phase separation. These regions include both 'driving' and 'participating' segments, as defined in *Figure 4* of the main text. *Figure 4—figure supplement 6* shows the distribution of $p$ across all proteins. For comparison, we also present the distribution of $1 - p$, which reflects the probability of finding conserved residues in non-phase-separating regions. While the average value of $p$ exceeds 0.5, indicating a trend toward conserved residues being more frequently located in phase-separating regions, the difference between the two distributions is not statistically significant. Future studies with expanded datasets may be necessary to clarify this trend.

Related work. Numerous studies have sought to identify functionally relevant amino acid groups within IDRs (*Wang et al., 2018*; *Li et al., 2024*; *Latham and Zhang, 2021*; *Zhang et al., 2021*; *Saar et al., 2021*; *Liu et al., 2025*; *Schuster et al., 2020*; *Pak et al., 2016*; *Cohan et al., 2022*; *Dao et al., 2018*). For instance, using multiple sequence alignment, several groups have identified evolutionarily conserved residues that contribute to phase separation (*Dasmeh et al., 2022*; *Ho and Huang, 2022*; *Jiang et al., 2015*; *Tatavosian et al., 2019*; *Schuster et al., 2020*; *Yang et al., 2019*; *Larson et al., 2017*; *DiRusso et al., 2023*; *Mitrea et al., 2016*; *Xiao et al., 2019*). Alderson et al. employed Alpha-Fold2 to detect disordered regions with a propensity to adopt structured conformations, suggesting potential functional relevance (*Alderson et al., 2023*).

In contrast, our approach based on ESM2 is more direct: it identifies conserved residues without relying on alignment or presupposing that functional significance requires folding into stable 3D structures. Notably, many of the conserved residues identified in our analysis exhibit low pLDDT scores (*Figure 2*), implying potential functional roles independent of stable conformations.

Relationship to the stickers-and-spacers model. To interpret the functional implications of conserved residues, we contextualize our findings within the stickers-and-spacers framework. Our results indicate that these conserved amino acids encompass both 'sticker' and 'spacer' classifications, as defined in recent literature (*Jiang et al., 2015*; *Schuster et al., 2020*; *Yang et al., 2019*; *Tatavosian et al., 2019*; *Larson et al., 2017*; *DiRusso et al., 2023*; *Mitrea et al., 2018*; *Xiao et al., 2019*; *Wang et al., 2021*; *Rekhi et al., 2024*). This suggests that residue-level classification may obscure functional roles in MLO formation.

Instead, evolutionarily conserved motifs, which can be readily identified through ESM2 score profiles, likely represent functionally significant units that integrate combinations of stickers and spacers. Perturbing these motifs in in vivo systems may offer a productive strategy for elucidating their biological functions.

Outside the conserved motifs, IDRs display greater mutational tolerance, with a general preference for hydrophilic residues. This structural arrangement—functionally constrained motifs embedded in more flexible, mutation-tolerant regions—is consistent with a generalized stickers-and-spacers framework. Conserved motifs likely engage in stronger interactions that drive phase separation, while interactions among non-motif regions remain attenuated. This spatial organization supports the formation of MLOs with defined chemical compositions and interaction specificity, as proposed by *Sood and Zhang, 2024*.

Future directions. Several promising directions could extend this work, both to refine our mechanistic understanding and to explore clinical relevance. One avenue is testing the hypothesis that conserved motifs in scaffold proteins act as functional stickers, mediating strong intermolecular interactions. This could be evaluated computationally via free energy calculations or experimentally via interaction assays. Deletion of such motifs in client proteins may also reduce their partitioning into condensates, illuminating their roles in molecular recruitment.

To explore potential clinical implications, we analyzed pathogenicity data from ClinVar (*Landrum et al., 2014*; *Landrum et al., 2018*; *Cagiada et al., 2024*; *Luppino et al., 2025*; *Fawzy and Marsh, 2025*). As shown in *Figure 5—figure supplement 3A*, single-point mutations with low LLR values—indicative of constrained residues—are enriched among clinically reported pathogenic variants, while benign variants typically exhibit higher LLR values. Moreover, mutations within conserved motifs are significantly more likely to be pathogenic than those in non-motif regions (*Figure 5—figure supplement 3B*). These findings highlight the potential of ESM2 as a first-pass screening tool for identifying clinically relevant residues and suggest that the conserved motifs described here may serve as priorities for future studies, both mechanistic and therapeutic.

Limitations. Despite these promising findings, our study has several limitations. Most notably, the analysis is entirely computational, relying on ESM2-derived predictions and sequence-based

conservation without direct experimental validation. While the strong correlation between ESM2 scores and evolutionary conservation suggests functional constraint, the specific biological roles of many conserved motifs remain uncharacterized.

The motifs we identify are located within broader regions that have previously been shown, through experimental studies, to drive phase separation. This provides circumstantial support for their functional relevance. However, these motifs typically represent only a small fraction of the larger disordered segments implicated in phase separation. As such, it is not possible to attribute the functional effect of the entire region solely to the identified motifs—other segments of the sequence may also make essential contributions. Therefore, while suggestive, the current evidence does not constitute direct experimental validation of motif-level functionality.

Future studies should aim to dissect the roles of these motifs in isolation and in context, using targeted mutagenesis, biophysical assays, and cellular perturbations. Such work will be crucial for establishing whether the conserved motifs identified here act as discrete functional units or as components of a broader sequence context required for biomolecular condensation and related processes.

## Methods
### Data collection and preprocessing
Human protein dataset. To construct the MLO-hProt dataset, we initially identified human proteins with disordered residues using the UniProtKB database (*Coudert et al., 2023*). From these proteins, we selected candidates with at least 10% of their residues exhibiting a pLDDT score ≤70, yielding a total of 5121 candidates. These were then cross-referenced with entries in the CD-code Database (*Rostam et al., 2023*), yielding a final subset of 939 proteins associated with MLO formation (*Figure 1A*). The remaining 4182 proteins, which were not linked to MLO formation, comprise the nMLO-hProt dataset. We further identified disordered regions from the datasets by selecting residues with pLDDT score ≤70, referred to as MLO-hIDR and nMLO-hIDR. The CD-code Database also classifies the MLO-hProt proteins into two categories: drivers ($n = 82$) and clients ($n = 814$), with the corresponding disordered regions labeled as dMLO-hIDR and cMLO-hIDR. Additional information on MLO-hIDR candidate proteins and their biological roles is available in (MLO-hIDR.csv).

dMLO-IDR dataset. In addition to datasets comprising only human proteins, we developed a specialized dMLO-IDR database incorporating proteins from diverse species involved in driving phase separation. This database includes all Driver proteins cataloged in the MLOsMetaDB database (*Orti et al., 2024*), which documents experimentally validated disordered and phase-separating regions. Beginning with 780 Driver proteins, we filtered the dataset to retain entries where disordered regions overlap with phase-separating segments, yielding 399 candidates across 40 species (*Figure 4A*). To further refine the dataset, we analyzed sequence identity, excluding homologous pairs with sequence identity exceeding 50%. This process resulted in a final set of 341 non-redundant candidates. These proteins play critical roles in mediating the formation of various MLOs, including P-bodies, stress granules, paraspeckles, and centrosomes (see *Supplementary file 3* (dMLO-IDR.csv)).

### AlphaFold2 score for structural order
The pLDDT scores for human proteins were retrieved from the AlphaFold Protein Structure Database (*Varadi et al., 2024*) by selecting the *Homo sapiens* organisms (Reference Proteome ID: UP000005640).

### ESM2 predictions for mutational preferences
We employed the code and pretrained parameters for ESM2 available from the model's official GitHub repository at https://github.com/facebookresearch/esm (*Hsu et al., 2023*) to conduct mutational predictions. To optimize computational efficiency, we utilized the esm2_t33_650M_UR50D model, which has 650 million parameters and achieves prediction accuracy comparable to the larger 15B parameter model (*Brandes et al., 2023*). In addition to calculating the LLR for individual mutations, we defined an ESM2 score at each position to quantify mutational tolerance, formulated as

$$\text{ESM2 Score} = -\sum_{i=1}^{20} \left( \frac{e^{\text{LLR}_i}}{\sum_{j=1}^{20} e^{\text{LLR}_j}} \log \left( \frac{e^{\text{LLR}_i}}{\sum_{j=1}^{20} e^{\text{LLR}_j}} \right) \right) \tag{1}$$

where $\text{LLR}_i$ denotes the LLR value for the ith amino acid mutation type.

## Evolutionary sequence analysis

We performed MSA analysis using HHblits from the HH-suite3 software suite (*Suzek et al., 2007*; *Suzek et al., 2015*; *Remmert et al., 2012*; *Steinegger et al., 2019*), a widely used open-source toolkit known for its sensitivity in detecting sequence similarities and identifying protein folds. HHblits builds MSAs through iterative database searches, sequentially incorporating matched sequences into the query MSA with each iteration. Sequence alignment was performed using the full-length protein sequences, encompassing both folded and disordered regions.

The HH-suite3 software was obtained from its GitHub repository (https://github.com/soedinglab/hh-suite; *Soeding et al., 2020*). Homologous sequences were identified through searches against the UniRef30 protein database (release 2023/02) (*Mirdita et al., 2017*). For each query, we performed three iterations of HHblits searches, incorporating sequences from profile HMM matches with an *E*-value threshold of 0.001 into the query MSA in each cycle. Using a lower *E*-value threshold (closer to 0) ensures greater sequence similarity among the matches, while multiple iterations enhance the alignment's depth and accuracy. The resulting alignments in A3M format were converted to CLUSTAL format using the reformat.pl script provided in HH-suite, aligning all sequences to a uniform length (*Figure 3A*).

To refine alignment quality by focusing on closely related homologs, we filtered out sequences with ≤20% identity to the query, excluding weakly related sequences where only short segments show similarity to the reference. For each sequence, we calculated the percent identity by counting the number of positions, denoted as $n$, at which the amino acid matches the reference. The percent identity was then computed as $n/N$, where $N$ represents the total length of the aligned reference sequence. This total includes residues in folded and disordered regions, as well as gap positions introduced during alignment.

The conservation score $CS_i$ for position $i$ was then calculated from the MSA as

$$CS_i = \frac{n_i(\text{ref=query})}{N_i(\text{non\_gap})}, \tag{2}$$

where $n_i(\text{ref=query})$ represents the number of times the residues from the reference sequence appear across all sequences, and $N_i(\text{non\_gap})$ represents the total non-gap residues across the aligned sequences.

## Motif identification

We defined motifs as contiguous stretches of amino acid sequences with an average ESM2 score of 0.5 or lower. To identify motifs within a given IDR, we implemented the following iterative procedure. Starting from either the N- or C-terminus of the sequence, we first locate the initial residue $i$ whose ESM2 score falls within 0.5. From $i$, residues are sequentially appended in the direction toward the opposite terminus until the segment's average ESM2 score exceeds 0.5; the first residue to breach this threshold is denoted $j$. The segment $(i, i + 1, \ldots, j - 1)$ is then recorded as a candidate motif. This process repeats starting from $j$ until the end of the IDR is reached.

We perform this full procedure independently from both termini and designate the final motif as the intersection of the two candidate-motif sets. This bidirectional overlap strategy excludes terminal residues that might transiently satisfy the average-score criterion only due to adjacent low-scoring regions, thereby isolating the conserved core of each motif. All other residues—those not included in either directional pass—are classified as non-motif regions, minimizing peripheral artifacts.

When two motifs are in close proximity along the sequence, they may be merged into a single motif. Specifically, if the starting position of one motif is within eight residues of the ending position of another, we define a candidate segment as the sequence spanning both motifs and the intervening residues. If the candidate segment's average ESM2 score is below 0.5, it is included as a merged motif, replacing the individual motifs in the final list (*Supplementary file 1* (ESM2_motif_with_exp_ref.csv)). In the analyses shown in *Figure 5*, we showed all motifs with $n \geq 4$; however, varying motif minimal length $n$ does not alter the overall conclusions (*Figure 5—figure supplement 2B*).

# Additional information

## Competing interests

Bin Zhang: Reviewing editor, eLife. The other authors declare that no competing interests exist.

## Funding

| Funder | Grant reference number | Author |
| --- | --- | --- |
| National Institute of General Medical Sciences | R35GM133580 | Bin Zhang |

The funders had no role in study design, data collection, and interpretation, or the decision to submit the work for publication.

## Author contributions

Yumeng Zhang, Conceptualization, Resources, Data curation, Software, Formal analysis, Validation, Investigation, Visualization, Methodology, Writing – original draft, Writing – review and editing; Jared Zheng, Software, Methodology, Writing – original draft, Writing – review and editing; Bin Zhang, Conceptualization, Data curation, Formal analysis, Supervision, Funding acquisition, Investigation, Visualization, Methodology, Writing – original draft, Project administration, Writing – review and editing

## Author ORCIDs

Yumeng Zhang (ID) https://orcid.org/0000-0002-6405-8362
Jared Zheng (ID) https://orcid.org/0009-0005-9653-8028
Bin Zhang (ID) https://orcid.org/0000-0002-3685-7503

Reviewer #1 (Public review): https://doi.org/10.7554/eLife.105309.3.sa1
Reviewer #2 (Public review): https://doi.org/10.7554/eLife.105309.3.sa2
Author response https://doi.org/10.7554/eLife.105309.3.sa3

# Additional files

## Supplementary files

MDAR checklist

Supplementary file 1. dMLO-IDR motifs identified by low ESM2 fitness scores and corroborated by experimental evidence.

Supplementary file 2. Protein-level annotations from the MLO-hIDR dataset, including UniProt identifiers, organism, database source, LLPS roles, associated membraneless organelles (MLOs), intrinsically disordered regions (IDRs), LCRs, LLPS regions, and Pfam domains.

Supplementary file 3. Protein-level annotations from the dMLO-IDR dataset, including UniProt identifiers, organism, database source, LLPS roles, associated membraneless organelles (MLOs), intrinsically disordered regions (IDRs), LCRs, LLPS regions, and Pfam domains.

## Data availability

The current manuscript is a computational study, so no data have been generated for this manuscript. Modeling code is uploaded at https://github.com/ZhangGroup-MITChemistry/ESM2_IDR_LLPS (copy archived at *Zhang, 2025*).

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
