## [Editor Report · eLife Assessment]

This **valuable** study presents an analysis of evolutionary conservation in intrinsically disordered regions, identified as key drivers of phase separation, leveraging a protein language model. The strength of evidence presented is **convincing** overall, though the theoretical grounding could benefit from further development.

---

## [Referee Report · Reviewer #1 (Public review)]

The manuscript by Zhang et al describes the use of a protein language model (pLM) to analyse disordered regions in proteins, with a focus on those that may be important in biological phase separation. This is an interesting study that supports, complements and extends previous related analyses on the conservation and mutational tolerance of disordered regions, with a particular focus on disordered regions in proteins that are found in condensates.

---

## [Referee Report · Reviewer #2 (Public review)]

This manuscript uses the ESM2 language model to map the evolutionary fitness landscape of intrinsically disordered regions (IDRs). The central idea is that mutational preferences predicted by these models could be useful in understanding eventual IDR-related behavior, such as disruption of otherwise stable phases. While ESM2-type models have been applied to analyze such mutational effects in folded proteins, they have not been used or verified for studying IDRs. Here, the authors use ESM2 to study membraneless organelle formation and the related fitness landscape of IDRs.

Through this, their key finding in this work is the identification of a subset of amino acids that exhibit mutation resistance. Their findings reveal a strong correlation between ESM2 scores and conservation scores, which if true, could be useful for understanding IDRs in general. Through their ESM2-based calculations, the authors conclude that IDRs crucial for phase separation frequently contain conserved sequence motifs composed of both so-called sticker and spacer residues. The authors note that many such motifs have been experimentally validated as essential for phase separation.

Comments on revisions:

Unfortunately my concerns about lack of theoretical grounding and validation (especially critical in lack of theoretical grounding) persist. The argument about correlation between ESM2 scores and MSA conservation is circular. Protein language models already encode residue‑level conservation, so agreement with conservation does not establish new predictive power. For IDRs, conservation is a poor surrogate for function because many functions are mediated by short, degenerate SLiMs that are frequently gained and lost. Sequence‑only predictions therefore need orthogonal (preferably experimental or at the least in silico) tests. Finally, without a family‑level holdout (e.g., cluster de‑duplication at low identity) and prospective tests, overlap with known motifs cannot rule out training‑data memorization/near‑duplicates.

---

## [Author Response]

The following is the authors’ response to the original reviews

**Reviewer #1:**
Summary:The manuscript by Zhang et al describes the use of a protein language model (pLM) to analyse disordered regions in proteins, with a focus on those that may be important in biological phase separation. While the paper is relatively easy to read overall, my main comment is that the authors could perhaps make it clearer which observations are new, and which support previous work using related approaches. Further, while the link to phase separation is interesting, it is not completely clear which data supports the statements made, and this could also be made clearer.

We thank the reviewer for their thoughtful evaluation of our manuscript and for the supportive comments. As outlined in the responses below, we have made substantial revisions to clarify the novel observations presented in our study and to strengthen the connection between sequence conservation and phase separation.

Comment 1: With respect to putting the work in a better context of what has previously been done before, this is not to say that there is not new information in it, but what the authors do is somewhat closely related to work by others. I think it would be useful to make those links more directly.

We have addressed the specific comments as outlined below.

Comment 1a: Alderson et al (reference 71) analysed in detail the conservation of IDRs (via pLDDT, which is itself related to conservation) to show, for example, that conserved residues fold upon binding. This analysis is very similar to the analysis used in the current study (using ESM2 as a different measure of conservation). Thus, the result that "Given that low ESM2 scores generally reflect mutational constraint in folded proteins, the presence of region a among disordered residues suggests that certain disordered amino acids are evolutionarily conserved and likely functionally significant" is in some ways very similar to the results of that (Alderson et al) paper .

We thank the reviewer for the comment. However, we would like to clarify that our findings show subtle but important differences from those reported by Alderson et al. Specifically, Alderson et al. used AlphaFold2 predictions to identify IDRs that undergo disorder-to-order transitions, which the authors termed as conditionally folded IDRs. These regions could potentially be functionally important, assuming that function of IDRs necessitate folding.

We argue, however, that, the validity of this structure-function relationship for IDRs remains to be tested. In our opinion, The most direct way to evaluate the functional significance is via evaluating the evolutionary conservation.

As shown in Author response image 1, the correlation between pLDDT scores and the conservation score, while noticable, is significantly weaker than that between the ESM2 score and the conservation score.

**Author response image 1. sa3fig1:** Comparison of the correlation between AlphaFold2 pLDDT scores and conservation scores with the correlation between ESM2 scores and conservation scores. Calculations were performed using proteins in the MLO-hProt dataset. (A) Correlation between the mean AlphaFold2 pLDDT scores and conservation scores for various amino acids. Pearson correlation coefficients (*r*) are indicated in the figure legends. The four panels on the right present analogous correlation plots for amino acids grouped by structural order, as defined by their pLDDT scores. (B) Similar as in part A but for ESM2 scores.

Therefore, we believe that ESM2 score is a better indicator than AlphaFold2 pLDDT score for functional relevance.

Furthermore, for the human IDRs, we explicitly selected amino acids with pLDDT scores ≤ 70.

These would be classified as structureless, disordered amino acids, according to the study by Alderson et al. Nevertheless, as shown in Figures 2 and 3 of the main text, our analyses still identifies conserved regions. Therefore, these regions may function via distinct mechanisms than the disorder to order transition.

We now discuss the novelty of our work in the context of existing studies in the newly added Conclusions and Discussion: Related Work, as quoted below.

“Numerous studies have sought to identify functionally relevant amino acid groups within IDRs [cite]. For instance, using multiple sequence alignment, several groups have identified evolutionarily conserved residues that contribute to phase separation [cite]. Alderson et al. employed AlphaFold2 to detect disordered regions with a propensity to adopt structured conformations, suggesting potential functional relevance [alderson et al].

In contrast, our approach based on ESM2 is more direct: it identifies conserved residues without relying on alignment or presupposing that functional significance requires folding into stable 3D structures. Notably, many of the conserved residues identified in our analysis exhibit low pLDDT scores (Figure 2), implying potential functional roles independent of stable conformations.”

Comment 1b: Dasmeh et al, Lu et al and Ho & Huang analysed conservation in IDRs, including aromatic residues and their role in phase separation.

We thank the reviewer for bringing these works to our attention! We now explicitly discuss these studies in both the Discussion section as mentioned above and in the Introduction as quoted below.

“Evolutionary analysis of IDRs is challenging due to difficulties in sequence alignment [cite], though several studies have attempted alignment of disordered proteins with promising results [Dasmeh et al, Lu et al and Ho & Huang].”

Comment 1c: A number of groups have performed proteomewide saturation scans using pLMs, including variants of the ESM family, including Meier (reference 89, but cited about something else) and Cagiada et al (https://doi.org/10.1101/2024.05.21.595203) that analysed variant effects in IDRs using a pLM. Thus, I think statements such as "their applicability to studying the fitness and evolutionary pressures on IDRs has yet to be established" should possibly be qualified.

We added a new paragraph in the *Introduction* to discuss the application of protein language models to IDRs and cited the suggested references.

“While protein language models have been widely applied to structured proteins [cite], it is important to emphasize that these models themselves are not inherently biased toward folded domains. For example, the Evolutionary Scale Model (ESM2) [cite] is trained as a probabilistic language model on raw protein sequences, without incorporating any structural or functional annotations. Its unsupervised learning paradigm enables ESM2 to capture statistical patterns of residue usage and evolutionary constraints without relying on explicit structural information. Thus, the success of ESM2 in modeling the mutational landscapes of folded proteins [cite] reflects the model’s ability to learn sequence-level constraints imposed by natural selection — a property that is equally applicable to IDRs if those regions are also under functional selection. Indeed, protein language models are increasingly been used to analyze variant effects in IDRs [cite].”

Comment 2: On page 4, the authors write, "The conserved residues are primarily located in regions associated with phase separation." These results are presented as a central part of the work, but it is not completely clear what the evidence is.

We thank the reviewer this insightful comment. We realized that our wording is not as precise as we should have been. We meant to state that the regions associated with phase separation are significantly enriched in these conserved residues. This is a significant finding and indicates that phase separation could be a source of evolutionary pressure in dictating IDP sequence conservation. However, we do not intend to suggest that phase separation is the only evolutionary pressure.

The sentence has been revised to

“Notably, regions associated with phase separation are significantly enriched in these conserved residues.”

We further replaced the section title "Conserved, Disordered Residues Localize in Regions Driving Phase Separation" with "Regions Driving Phase Separation Are Enriched with Conserved, Disordered Residues" to further clarify our findings and avoid overinterpretation.

Finally, we revised the following sentence in the discussion

“Notably, these conserved, disordered residues are predominantly located in regions actively involved in phase separation, contributing to the formation of membraneless organelles.”

to

“Notably, regions actively involved in phase separation are enriched with these conserved, disordered residues, supporting their potential role in the formation of membraneless organelles.”

The submitted manuscript provides clear evidence supporting the enrichment of conserved residues in MLO-driving IDRs. Specifically, Figures 4A and 4C demonstrate that these IDRs exhibit a substantially higher fraction of conserved residues compared to other IDRs involved in phase separation.

In this analysis, the nMLO-hIDR group serves as a baseline, representing the distribution of conservation in disordered regions lacking MLO-related functions. In contrast, IDRs from MLOassociated groups show a pronounced lower shift in their median and interquartile ranges, indicating stronger evolutionary constraints. Within the dMLO cohort, the degree of conservation follows a distinct gradient: driving residues exhibit the highest levels of conservation, followed by participant residues, with non-participant residues showing values closer to the nMLO baseline. This pattern reflects the relative functional importance of each group in phase separation, with conservation levels corresponding to their roles in MLO scaffolding.

To further support this, we computed, for each IDR, the fraction of conserved amino acids. As shown in Figure S11B, for IDRs that actively contribute to phase separation, the fraction is indeed higher than those not involved in phase separation. This analysis is now included in SI.

During the revision, we explicitly evaluated whether conserved residues are preferentially located in regions associated with phase separation. To this end, for each protein in the MLO-hProt dataset, we calculated the probability *p* of finding conserved residues within regions contributing to phase separation. These regions include both "driving" and "participating" segments as defined in Figure 4 of the main text.

Figure S11A presents the distribution of *p* across all proteins. For comparison, we also include the distribution of 1− *p*, representing the probability of finding conserved residues in regions not associated with phase separation. On average, *p* exceeds 0.5, suggesting a tendency for conserved residues to be more frequently located in phase-separating regions. However, the difference between the two distributions is not statistically significant. This result may be due to the generally low density of conserved residues in IDRs, which makes the estimation of *p* challenging for individual proteins. Additionally, some conserved sites may be involved in functions unrelated to phase separation.

We added the following text to the *Discussion* section of the main text.

“We emphasize that the results presented in Figure 4 do not directly demonstrate that conserved residues are preferentially located in regions associated with phase separation. Although these regions are more enriched in conserved amino acids, their total sequence length can be smaller than that of non-phase-separating regions. As a result, the absolute number of conserved residues may still be higher outside phase-separating regions. To quantitatively assess this, we calculated, for each protein in the MLO-hProt dataset, the probability *p* of finding conserved residues within regions contributing to phase separation. These regions include both "driving" and "participating" segments, as defined in Figure 4 of the main text. Figure S11 shows the distribution of *p* across all proteins. For comparison, we also present the distribution of 1− *p*, which reflects the probability of finding conserved residues in non-phase-separating regions. While the average value of *p* exceeds 0.5, indicating a trend toward conserved residues being more frequently located in phase-separating regions, the difference between the two distributions is not statistically significant. Future studies with expanded datasets may be necessary to clarify this trend.”

Comment 3: It would be useful with an assessment of what controls the authors used to assess whether there are folded domains within their set of IDRs.

We acknowledge that our previous labeling may have caused some confusion. Protein sequences used in Figures 2 and 3 include both folded and disordered domains. Results presented in these figures were constructed using full-length protein sequences to highlight the similarities and differences in ESM2 scores between folded and disordered domains.

In contrast, the analyses presented in Figures 4 and 5 focus exclusively on IDRs to examine their role in phase separation.

To prevent further confusion, we have renamed the dataset used in Figures 2 and 3 as MLO-hProt, emphasizing that the analysis pertains to entire protein sequences. The term MLO-hIDR is now reserved for a new dataset that includes only disordered residues, as used in Figures 4 and 5, and corresponding SI Figures.

For the dMLO-IDR dataset, all except one amino acid (P40967, residue G592) are annotated as disordered in the MobiDB database (https://mobidb.org/). This database characterizes disordered regions based on a combination of predictive algorithms and experimental data. As illustrated in Figure S5A, 25.5% of the proteins in the dataset have direct experimental evidence supporting their disorderedness. These experimental annotations are derived from a diverse range of techniques (Figure S5B). For the remaining proteins, disorder was predicted by one or more computational tools. Although not all tools were applied to every protein, each protein in the dataset was identified as disordered by at least one method.

For human proteins, IDRs were identified based on AlphaFold2 pLDDT scores, using a threshold of 70. As established in prior studies [1, 2], the pLDDT score provides a quantitative measure of local structural confidence, with lower values indicating greater structural disorder. IDRs associated with conditional folding or disorder-to-order transitions generally exhibit high pLDDT values (e.g., >70).

Author response image 2 shows a violin plot of AlphaFold2 pLDDT scores for the various MLO-hIDR groups. The consistently low scores support the conclusion that these regions are structurally disordered.

We also cross-checked the MLO-hIDR regions against the MobiDB database. As shown in Figure S6, approximately 76% of the proteins in the dataset are predicted to contain disordered regions. Among the non-labeled segments with pLDDT scores ≤ 70, the majority are relatively short, with segments of 1–5 residues accounting for approximately 80%.

**Author response image 2. sa3fig2:** AlphaFold pLDDT scores of hIDRs in different MLO-related groups.

In addition to renaming the dataset, we also revised the manuscript to highlight the validation of disorderedness in section of *Results: Regions Driving Phase Separation Are Enriched with Conserved, Disordered Residues*.

“The presence of evolutionarily conserved disordered residues raises the question of their functional significance. To explore this, we identified disordered regions of MLO-hProt using a pLDDT score less than 70 and partitioned these regions into two categories: drivers (dMLO-hIDR), which actively drive phase separation, and clients (cMLO-hIDR), which are present in MLOs under certain conditions but do not promote phase separation themselves [cite]. Additionally, IDRs from human proteins not associated with MLOs, termed nMLO-hIDR, were included as a control. To enhance statistical robustness, we extended our dataset by incorporating driver proteins from additional species [cite], resulting in the expanded dMLO-IDR dataset. Beyond the pLDDT-based classification, the majority of residues in these datasets are also predicted to be disordered by various computational tools and supported by experimental evidence (Figures S5 and S6).”

Recommendation 1: The authors use the terms "evolutionary fitness of IDRs" (abstract and p. 5, for example), "fitness of amino acids" (p. 4), and "quantify the fitness of particular residues at specific sites" (p. 5). It is not clear what is meant by fitness in this context.

We thank the reviewer for pointing out the ambiguity in the term fitness. To enhance clarity, we have replaced “fitness" with “mutational tolerance" to more directly emphasize the evolutionary conservation of specific residues.

Recommendation 2: The authors write (p. 6) "Previous studies have demonstrated a strong correlation between ESM2 scores and changes in free energy related to protein structure stability". While that may be true, it might be worth noting that ESM2 scores report on the effects of mutations and function more broadly than stability because these models have previously been shown to capture conservation effects beyond stability.

We fully agree with the reviewer’s comment and have revised the main text accordingly. Specifically, the referenced sentence has been revised and relocated, as shown below.

“Our analysis demonstrated that HP1_α_’s structured domains consistently yield low ESM2 scores, reflecting strong mutational constraints characteristic of folded regions. These constraints are further evident in the local LLR predictions, as shown in Figure 2B, where we illustrate the folded region G120-T130. Given the functional importance of preserving the 3D of structured domains, mutations with greater detrimental effects are likely to disrupt protein folding substantially. This interpretation is consistent with previous studies reporting a significant correlation between ESM2 LLRs and changes in free energy associated with protein structural stability [cite].”

Recommendation 3: p. 10: The authors write "To exclude sequences that no longer qualify as homologs, we filtered for sequences with at least 20% identity to the reference". How did they decide on 20% and why? And over which residues are these 20% calculated.

We apologize for the earlier lack of clarity. Sequence alignment was performed using the full-length protein sequences, encompassing both folded and disordered regions. For each sequence, we calculated the percent identity by counting the number of positions, denoted as *n*, at which the amino acid matches the reference. The percent identity was then computed as *n*/*N*, where *N* represents the total length of the aligned reference sequence. This total includes residues in folded and disordered regions, as well as gap positions introduced during alignment.

We updated the Methods section of the main text to clarify.

“We performed multi-sequence alignment (MSA) analysis using HHblits from the HH-suite3 software suite [citations], a widely used open-source toolkit known for its sensitivity in detecting sequence similarities and identifying protein folds. HHblits builds MSAs through iterative database searches, sequentially incorporating matched sequences into the query MSA with each iteration. Sequence alignment was performed using the full-length protein sequences, encompassing both folded and disordered regions.

...

To refine alignment quality by focusing on closely related homologs, we filtered out sequences with ≤ 20% identity to the query, excluding weakly related sequences where only short segments show similarity to the reference. For each sequence, we calculated the percent identity by counting the number of positions, denoted as *n*, at which the amino acid matches the reference. The percent identity was then computed as *n*/*N*, where *N* represents the total length of the aligned reference sequence. This total includes residues in folded and disordered regions, as well as gap positions introduced during alignment.”

We selected a 20% sequence identity threshold to balance inclusion of true homologs with exclusion of distant matches that may not share functional relevance. To determine this cutoff, we compared identity thresholds of 0%, 10%, 20%, and 40% and examined the resulting distributions of conservation and ESM2 scores across aligned residues for MLO-hProt dataset (Author response image 3). Thresholds of 10%, 20%, and 40% produced qualitatively similar results, with a consistent correspondence between low ESM2 scores and high conservation. Lower thresholds introduced highly divergent sequences that added noise to the alignment, resulting in reduced overall conservation scores. In contrast, higher thresholds excluded homologs with potentially meaningful conservation, particularly in disordered regions where conservation scores tend to be relatively low.

**Author response image 3. sa3fig3:** Histograms of the ESM2 score and the conservation score, presented in a format consistent with Figure 3B of the main text. The conservation scores were computed using aligned sequences with identity thresholds of ≥0, ≥10%, ≥20%, and ≥40% (left to right). Contour lines represent different levels of −log_P_(CS,ESM2), where *P* is the joint probability density of conservation score (CS) and ESM2 score. Contours are spaced at 0.5-unit intervals, highlighting regions of distinct density.

Recommendation 4: In their description of "motif" searching algorithm (p. 20) I think that the search algorithm would give a different result whether the search is performed N->C or C->N because the first residue (i) needs to have a score <0.5 but the last (j) could have a score > 0.5 as long as the average is below 0.5. Is that correct? And if so, why did they choose an asymmetric algorithm?

We thank the reviewer for highlighting the asymmetry in our motif-search algorithm.

To investigate this issue, we repeated the algorithm starting from the C-terminus and compared the resulting motifs with those obtained from the N-terminal scan. We found that the two sets of motifs overlap entirely: each motif identified from the C-terminal direction has a corresponding counterpart from the N-terminal scan. However, the motifs are not identical. The directionality of the search introduces additional amino acids—referred to here as peripheral residues—at the motif boundaries, which differ between the two sets.

As shown in Author response image 4, the number of peripheral residues is small relative to the total motif length.

To eliminate asymmetry and ambiguity, we have revised our method to perform bidirectional scans—from both the N- and C-termini—and define each motif as the overlapping region identified by both directions. This approach emphasizes the conserved core and avoids the inclusion of spurious terminal residues. The updated procedure is described in Methods: Motif Identification.

“To identify motifs within a given IDR, we implemented the following iterative procedure. Starting from either the N– or C–terminus of the sequence, we first locate the initial residue *i* whose ESM2 score falls within 0.5. From *i*, residues are sequentially appended…”

**Author response image 4. sa3fig4:** Number of peripheral residues and their relative length to the full-motif length identified from both sides. (A). The unique motifs identified from N-to-C terminal direction. (B) The unique motifs identified from C-to-N terminal direction.

“…in the direction toward the opposite terminus until the segment’s average ESM2 score exceeds 0.5; the first residue to breach this threshold is denoted *j*. The segment (*i*,*i*+1,*...*, *j*−1) is then recorded as a candidate motif. This process repeats starting from *j* until the end of the IDR is reached.

We perform this full procedure independently from both termini and designate the final motif as the intersection of the two candidate-motif sets. This bidirectional overlap strategy excludes terminal residues that might transiently satisfy the average-score criterion only due to adjacent low-scoring regions, thereby isolating the conserved core of each motif. All other residues—those not included in either directional pass—are classified as non-motif regions, minimizing peripheral artifacts.”

Accordingly, we have updated the *Supplementary material:* ESM2_motif_with_exp_ref.csv for the new identified motifs commonly exited from both N-terminal and C-terminal searches. Minor changes were observed in the set of motifs as being discussed, but these do not affect the main conclusions. Figures 5C, 5D, and S6 have been revised accordingly.

**Reviewer #2:**
Summary:Unfortunately, I do not believe that the results can be trusted. ESM2 has not been validated for IDRs through experiments. The authors themselves point out its little use in that context. In this study, they do not provide any further rationale for why this situation might have changed. Furthermore, they mention that experimental perturbations of the predicted motifs in in vivo studies may further elucidate their functional importance, but none of that is done here. That some of the motifs have been previously validated does not give any credibility to the use of ESM2 here, given that such systems were probably seen during the training of the model.

We thank the reviewer for their detailed and thoughtful critique of our manuscript. We recognize the importance of careful model validation, especially in the context of IDRs, and appreciate the opportunity to clarify the scope and rationale of our study. Below, we respond point-by-point to the main concerns.

(1) The use of ESM2 is not validated for IDRs, and the authors provide no rationale for its applicability in this context.

We thank the reviewer for raising this important point.

First, we emphasize that ESM2 is a probabilistic language model trained entirely on amino acid sequences, without any structural supervision. The model does not receive any input about protein structure — folded or disordered — during training. Instead, it learns to estimate the likelihood of each amino acid at a given position, conditioned on the surrounding sequence context. This makes ESM2 agnostic to whether a sequence is folded or disordered; the model’s capacity to identify patterns of residue usage arises solely from the statistics of natural sequences.

As such, ESM2 is not inherently biased toward folded proteins, even though previous studies have demonstrated its usefulness in identifying conserved and functionally constrained residues in structured domains [3–9]. These findings support the broader utility of language models for uncovering evolutionary constraints — and by extension, suggest that similar signatures could exist in IDRs, particularly if they are under functional selection.

Indeed, if certain residues or motifs in IDRs are conserved due to their importance in biological processes (e.g., phase separation), we would expect such selection to be reflected in sequence-based features, which ESM2 is designed to detect. The model’s applicability to IDRs, then, is a natural extension of its core probabilistic architecture.

To further evaluate this, we carried out an independent in silico validation using multiple sequence alignments (MSAs). This analysis allowed us to compute the evolutionary conservation of individual amino acids without any reliance on ESM2. We then compared these conservation scores to ESM2 scores and found a strong correlation between the two. This provides direct, quantitative support for the idea that ESM2 is capturing biologically meaningful sequence constraints — even in disordered regions.

While we agree that experimental testing would ultimately provide the most compelling validation, we believe that our MSA-based comparison constitutes a strong and arguably ideal computational validation of the model’s predictions. It offers an orthogonal measure of evolutionary pressure that confirms the biological plausibility of ESM2 scores.

We added the following text in the introduction to highlight the applicability of ESM2 to IDRs.

“While protein language models have been widely applied to structured proteins, it is important to emphasize that these models themselves are not inherently biased toward folded domains. For example, the Evolutionary Scale Model (ESM2) [cite] is trained as a probabilistic language model on raw protein sequences, without incorporating any structural or functional annotations. It operates by estimating the likelihood of observing a given amino acid at a particular position, conditioned on the entire surrounding sequence context. This unsupervised learning paradigm enables ESM2 to capture statistical patterns of residue usage and evolutionary constraints without relying on explicit structural information. Thus, the success of ESM2 in modeling fitness landscapes of folded proteins reflects the model’s ability to learn sequence-level constraints imposed by natural selection — a property that is equally applicable to IDRs if those regions are also under functional selection. Indeed, protein language models are increasingly been used to analyze variant effects in IDRs [cite].”

(2) There is no experimental validation of the ESM2-based predictions in this study.

We agree that experimental validation would provide definitive support for the utility of ESM2 in IDRs, and we explicitly state this as a limitation in the revised manuscript as quoted below.

“Limitations: Despite the promising findings, our study has several limitations. Most notably, our analysis is purely computational, relying on ESM2-derived predictions and sequence-based conservation without accompanying experimental validation. While the strong correlation between ESM2 scores and evolutionary conservation provides compelling evidence that the identified motifs are functionally constrained, the precise biological roles of these motifs remain uncharacterized. ESM2 is well-suited for highlighting regions under selective pressure, but it does not provide mechanistic insights into how conserved motifs contribute to specific molecular functions such as phase separation, molecular recognition, or dynamic regulation. Determining these roles will require targeted experimental investigations, including mutagenesis and biophysical characterization.”

In addition, we revised the manuscript title from “Protein Language Model Identifies Disordered, Conserved Motifs Driving Phase Separation" to “Protein Language Model Identifies Disordered, Conserved Motifs Implicated in Phase Separation". This revision softens the original claim to better reflect the absence of direct experimental evidence for the motifs’ role in phase separation.

However, we also emphasize that the goal of our study is not to claim definitive predictive power, but rather to explore whether ESM2-derived mutational profiles align with known biological features of IDRs — and in doing so, to generate new, testable hypotheses.

In addition, while no in vivo experiments were performed, our study does include an in silico validation step, as detailed in the response to the previous comment. The strong correlation between ESM2 scores and conservation scores provides direct support for the utility of ESM2 in identifying residues under evolutionary constraint in disordered regions.

(3) The overlap between predicted motifs and known ones may be due totraining data leakage.

We respectfully clarify that training data leakage is not possible in this case, as ESM2 is trained using unsupervised learning on raw protein sequences alone. The model has no access to experimental annotations, functional labels, or knowledge of which motifs are involved in phase separation. It only models statistical sequence patterns derived from evolutionarily observed proteins.

Therefore, any agreement between ESM2-derived predictions and previously validated motifs arises not from memorization of experimental data, but from the model’s ability to learn meaningful sequence constraints from the natural distribution of proteins.

(4) The authors should revamp the study with a testable predictive framework.

We respectfully suggest that a full revamp is not necessary or appropriate in this context.

As outlined in our previous responses, we believe that certain misunderstandings about the nature and capabilities of ESM2 may have influenced the reviewer’s assessment.

Importantly, both Reviewer 1 and Reviewer 3 express strong support for the significance and novelty of this work, and recommend publication following minor revisions.

In this context, we believe the manuscript provides a useful contribution as a first step toward understanding disordered regions using language models, and that it has value even in the absence of direct experimental testing. We have now better positioned the manuscript in this light, clarified limitations, and suggested concrete next steps for follow-up research.

We hope these clarifications and revisions address the reviewer’s concerns, and we thank them again for helping us strengthen the framing, rigor, and clarity of our study.

**Reviewer #3:**
Summary:This is a very nice and interesting paper to read about motif conservation in protein sequences and mainly in IDRs regions using the ESM2 language model. The topic of the paper is timely, with strong biological significance. The paper can be of great interest to the scientific community in the field of protein phase transitions and future applications using the ESM models. The ability of ESM2 to identify conserved motifs is crucial for disease prediction, as these regions may serve as potential drug targets. Therefore, I find these findings highly significant, and the authors strongly support them throughout the paper. The work motivates the scientific community towards further motif exploration related to diseases.Strengths:(1) Revealing conserved regions in IDRs by the ESM-2 language model.(2) Identification of functionally significant residues within protein sequences, especially in IDRs.(3) Findings supported by useful analyses.

We appreciate the reviewer’s thoughtful words and support for our work.

Weaknesses:(1) Lack of examples demonstrating the potential biological functions of these conserved regions.

As detailed in the Response to Recommendation 6, we conducted additional analyses to connect the identified conserved regions with their biological functions.

(2) Very limited discussion of potential future work and of limitations.

We have substantially revised the Conclusions and Discussion section to provide a detailed analysis of the study’s limitations and to propose several directions for future research, as elaborated in our Response to Recommendation 5 below.

Recommendation 1: The authors describe the ESM2 score such that lower scores are associated with conserved residues, stating that "lower scores indicate higher mutational constraint and reduced flexibility, implying that these residues are more likely essential for protein function, as they exhibit fewer permissible mutational states." However, when examining intrinsically disordered regions (IDRs), which are known to drive phase separation, I observe that the ESM2 score is relatively high (Figure 3C, pLDDT < 50, and Supplementary Figure S2). Could the authors clarify how this relatively high score aligns with the conservation of motifs that drive phase separation?

We thank the reviewer for this insightful comment. We would like to clarify that most amino acids in the IDRs are not conserved, even for IDRs that contribute to phase separation. Only a small set of amino acids in these IDRs, which we term as motifs, are evolutionarily conserved with low ESM2 scores. Therefore, the ESM2 scores exhibit bimodal distribution at high and low values, as shown in Figures 4A and 4C of the manuscript. When averaged over all the amino acids, the mean ESM2 scores, plotted in Figure 3C, are relatively high due to dominant population of non-conserved amino acids.

Recommendation 2: The authors mention: "We first analyzed the relationship between ESM2 and pLDDT scores for human Heterochromatin Protein 1 (HP1, residues 1-191)". I appreciate this example as a demonstration of amino acid conservation in IDRs. However, it is questionable whether the authors could provide some more examples to support amino acid conservation particularly within the IDRs along with lower ESM2 score (e.g, Could the authors provide some additional examples of "conserved disordered" regions in various proteins which are associated with relatively low ESM2 score as appear in Figure 2A).

We thank the reviewer for this valuable suggestion. We want to kindly noted that the conserved residues on IDRs are prevalent as indicated in Figures 2D and 3B. To further illustrate the prevalence of “conserved disordered” regions, we generated ESM2 versus pLDDT score plots for the full dMLO–hProt dataset (82 proteins) in Figure S2. In these plots, residues with pLDDT ≤ 70 are highlighted in blue to denote structural disorder (dMLO-hIDR), and these disordered residues with ESM2 score ≤ 1.5 are shown in purple to indicate conserved disordered segments.

Recommendation 3: Could the authors plot a Violin conservation score plot for Figure 4A to emphasise the relationship between ESM2 scores and conservation scores of disordered residues?

We thank the reviewer for this suggestion. We included a violin plot illustrating the distribution of conservation scores for disordered residues across all four IDR groups, shown in Author response image 5. Consistent with the findings in Figure 4A, the phase separation drivers (dMLO-hIDR and dMLOIDR) exhibit a higher proportion of conserved amino acids compared to the client group (cMLOhIDR).

We also note that the nMLO-hIDR group may contain conserved residues due to functions unrelated to MLO formation, which could contribute to the higher observed levels of conservation in this group.

**Author response image 5. sa3fig5:** Violin plots illustrating the distribution of conservation scores for disordered residues across the nMLO–hIDR, cMLO–hIDR, dMLO–hIDR, and dMLO–IDR datasets. Pairwise statistical comparisons were conducted using two-sided Mann–Whitney U tests on the conservation score distributions (null hypothesis: the two groups have equal medians). P-values indicate the probability of observing the observed rank differences under the null hypothesis. Statistical significance is denoted as follows: ***: p < 0.001; **: p < 0.01; *:p < 0.05;

Recommendation 4: It will be appreciated if the authors could add to Figure 4 Violin plots, a statistical comparison between the different groups.

We thank the reviewer for this valuable suggestion. We included the p-values for Figures 4A and 4C to quantify the statistical significance of differences in the distributions.

Most comparisons are highly significant (p < 0.001), while the largest p-value (p = 0.089) between the conservation score of driving and non-participating groups (Figure 4C) still suggests a marginally significant trend.

Recommendation 5: Could the authors expand more on potential future research directions using ESM2, given its usefulness in identifying conserved motifs? Specifically, how do the authors envision conserved motifs will contribute to future discoveries/applications/models using ESM (e.g, discuss the importance of conserved motifs, especially in IDRs motifs, in protein phase transition prediction in relation to diseases).

We thank the reviewer for this insightful comment. To further assess the functional relevance of the conserved motifs, we incorporated pathogenic variant data from ClinVar [10, 11] to evaluate mutational impacts. As shown in Figure S12A and B, a substantial number of pathogenic variants in MLO-hProt proteins are associated with low ESM2 LLR values. This pattern holds for both folded and disordered residues.

Moreover, we observed that variants located within motifs are more frequently pathogenic compared to those outside motifs (Figure S12C). In the main text, motifs were defined only for driver proteins; however, the available variant data for this subset are limited (6 data points). To improve statistical power, we extended motif identification to include both client and driver human proteins, following the same methodology described in the main text. Consistent with previous findings, variants within motifs in this expanded set are also more likely to be pathogenic. These results further support the functional importance of both low ESM2-scoring residues and the conserved motifs in which they reside.

The following text was added in the *Discussion* section of the manuscript to discuss these results and outline future research directions.

“Several promising directions could extend this work, both to refine our mechanistic understanding and to explore clinical relevance. One avenue is testing the hypothesis that conserved motifs in scaffold proteins act as functional stickers, mediating strong intermolecular interactions. This could be evaluated computationally via free energy calculations or experimentally via interaction assays. Deletion of such motifs in client proteins may also reduce their partitioning into condensates, illuminating their roles in molecular recruitment.

To explore potential clinical implications, we analyzed pathogenicity data from Clin-Var [10, 11]. As shown in Figure S12A, single-point mutations with low LLR values—indicative of constrained residues—are enriched among clinically reported pathogenic variants, while benign variants typically exhibit higher LLR values. Moreover, mutations within conserved motifs are significantly more likely to be pathogenic than those in non-motif regions (Figure S12B). These findings highlight the potential of ESM2 as a first-pass screening tool for identifying clinically relevant residues and suggest that the conserved motifs described here may serve as priorities for future studies, both mechanistic and therapeutic.”

Moreover, the functional significance of conserved motifs, particularly their implications in disease and pathology, warrants further investigation. As an initial analysis, we incorporated ClinVar pathogenic variant data [citation] to assess mutational effects within our datasets. As illustrated in Figure R12A, single-point mutations with low LLR values are enriched among clinically reported pathogenic variants, whereas benign variants are more commonly associated with higher LLR values. Notably, mutations within conserved motifs are substantially more likely to be pathogenic compared to those in non-motif regions. These findings highlight the potential of ESM2 as a firstpass tool for identifying residues of clinical relevance. The conserved motifs identified here may be prioritized in future studies aimed at elucidating their biological roles and evaluating their viability as therapeutic targets.

Recommendation 6: The authors mention: "Our findings provide strong evidence for evolutionary pressures acting on specific IDRs to preserve their roles in scaffolding phase separation mechanisms, emphasizing the functional importance of entire motifs rather than individual residues in MLO formation." They also present a word cloud of functional motifs in Figure 5D. Although it makes sense that evolutionarily conserved motifs, especially within the IDRs regions, act as functional units, I think there is no direct evidence for such functionality (e.g., examples of biological pathways associated with IDRs and phase separation). Hence, there is no justification to write in the figure caption: "ESM2 Identifies Functional Motifs in driving IDRs" unless the authors provide some examples of such functionality. This will even make the paper stronger by establishing a clear connection to biological pathways, and hence these motifs can serve as potential drug targets.

We thank the reviewer for this insightful suggestion. We have replaced “functional motifs" with “conserved motifs" in the figure caption.

Identifying the precise biological pathways associated with the conserved motifs is a complex task and a comprehensive investigation lies beyond the scope of this study. Nonetheless, as an initial effort, we explored the potential functions of these motifs using annotations available in DisProt (https://disprot.org/).

DisProt is the leading manually curated database dedicated to IDPs, providing both structural and functional annotations. Expert curators compile experimentally validated data, including definitions of disordered regions, associated functional terms, and supporting literature references. Author response image 6 presents a representative DisProt entry for DNA topoisomerase 1 (UniProt ID: P11387), illustrating its structural and biological annotation.

For each motif, we located the corresponding DisProt entry and assigned a functional annotation based on the annotated IDR from which the motif originates. We emphasize that this functional assignment should be regarded as an approximation. Because experimental annotations often pertain to the entire IDR, regions outside the motif may also contribute to the reported function.

Nevertheless, the annotations provide valuable insights.

**Author response image 6. sa3fig6:** Screenshot of information provided by the DisProt database. Detailed annotations of biological functions and structural features, along with experimental references, are accessible via mouse click.

Approximately 50% of ESM2-predicted IDR motifs lack functional annotations. Among those that are annotated, motifs from the dMLO-IDR dataset are predominantly associated with “molecular condensate scaffold activity,” followed by various biomolecular binding functions (Author response image 7A). These findings support the role of these motifs in MLO formation.

For comparison, we applied the same identification procedure (described in Methods: Motif Identification) to motifs from the nMLO-hIDR dataset. In contrast to the dMLO-IDR motifs, these exhibit a broader range of annotated functions related to diverse cellular processes. Collectively, these results suggest that motifs identified by ESM2 are aligned with biologically relevant functions captured in current databases.

Finally, as illustrated in Figure S12 and discussed in the Response to Recommendation 5, variants occurring within identified motifs are more likely to be pathogenic than those in non-motif regions, further underscoring their functional importance.

**Author response image 7. sa3fig7:** Biological functions of ESM2-predicted motifs. (A) Distribution of biological functions associated with all identified motifs from dMLO-IDR driving groups. (B) Distribution of biological functions associated with all identified motifs from nMLO-hIDR groups.

Recommendation 7: In Figure 2C the authors present FE (I assume this is free energy), some discussion about the difference in the free energy referring to the "a" region is missing i.e. both "Folded" and "Disordered" regions are associated with low ESM score but with low and high free energy (FE), respectively.

We thank the reviewer for the comments. FE indeed abbreviates free energy. To improve clarify and avoid confusion, we have updated all figure captions by replacing “FE” with “−log*P*” to explicitly denote the logarithm of probability in the contour density plots.

We used “a" in Figures 2C and 2D to refer to regions with low ESM2 scores, which appears a local minimum in both plots. Since most residues in folded regions are conserved, region a has lower free energy than region b in Figure 2C. On the other hand, as most residues in disordered regions are not conserved, as we elaborated in *Response to Recommendation 1*, region a has lower population and higher free energy than region b.

To avoid confusion, we have replaced “a" and “b" in Figure 2D with “I" and “II".

Recommendation 8: Figure S2: It would be useful to plot the same figure for structured and disordered regions as well.

We are not certain we fully understood this comment, as we believe the requested analysis has already been addressed. In Figure S2, we used the AlphaFold2 pLDDT score to represent the structural continuum of different protein regions, where residues with pLDDT > 70 (red and lightred bars) are classified as structured, while those with pLDDT ≤ 70 (blue and light-blue bars) are classified as disordered.

Minor suggestion 1: Could the authors clarify the meaning of the abbreviation "FE" in the colorbar of the contour line? I assume this is free energy.

We have updated all contour density plot figure captions by replacing “FE” with “−log*P*” to explicitly denote the logarithm of probability.

Minor suggestion 2: In Figure 2A - do the authors mean "Conserved folded" instead of just "Folded"? If so, could the authors indicate this?

We thank the reviewer for this comment. The ESM2 scores indeed suggest that, within folded regions, there may be multiple distinct groups exhibiting varying degrees of evolutionary conservation. However, as our primary focus is on IDRs, we chose not to investigate these distinctions further.

Figure 2A illustrates a randomly selected folded region based on AlphaFold2 pLDDT scores.

References

(1) Ruff, K. M.; Pappu, R. V. AlphaFold and Implications for Intrinsically Disordered Proteins. Journal of Molecular Biology 2021, 433, 167208.

(2) Alderson, T. R.; Pritišanac, I.; Kolaric, Ð.; Moses, A. M.; Forman-Kay, J. D. Systematic´ Identification of Conditionally Folded Intrinsically Disordered Regions by AlphaFold2. Proceedings of the National Academy of Sciences of the United States of America, 120, e2304302120.

(3) Brandes, N.; Goldman, G.; Wang, C. H.; Ye, C. J.; Ntranos, V. Genome-Wide Prediction of Disease Variant Effects with a Deep Protein Language Model. Nature Genetics 2023, 55, 1512–1522.

(4) Lin, Z. et al. Evolutionary-Scale Prediction of Atomic-Level Protein Structure with a Language Model. 2023.

(5) Zeng, W.; Dou, Y.; Pan, L.; Xu, L.; Peng, S. Improving Prediction Performance of General Protein Language Model by Domain-Adaptive Pretraining on DNA-binding Protein. Nature Communications 2024, 15, 7838.

(6) Gong, J. et al. THPLM: A Sequence-Based Deep Learning Framework for Protein Stability Changes Prediction upon Point Variations Using Pretrained Protein Language Model. Bioinformatics 2023, 39, btad646.

(7) Lin, W.; Wells, J.; Wang, Z.; Orengo, C.; Martin, A. C. R. Enhancing Missense Variant Pathogenicity Prediction with Protein Language Models Using VariPred. Scientific Reports 2024, 14, 8136.

(8) Saadat, A.; Fellay, J. Fine-Tuning the ESM2 Protein Language Model to Understand the Functional Impact of Missense Variants. Computational and Structural Biotechnology Journal 2025, 27, 2199–2207.

(9) Chu, S. K. S.; Narang, K.; Siegel, J. B. Protein Stability Prediction by Fine-Tuning a Protein Language Model on a Mega-Scale Dataset. PLOS Computational Biology 2024, 20, e1012248.

(10) Landrum, M. J.; Lee, J. M.; Riley, G. R.; Jang, W.; Rubinstein, W. S.; Church, D. M.; Maglott, D. R. ClinVar: Public Archive of Relationships among Sequence Variation and Human Phenotype. Nucleic Acids Research 2014, 42, D980–D985.

(11) Landrum, M. J. et al. ClinVar: Improving Access to Variant Interpretations and Supporting Evidence. Nucleic Acids Research 2018, 46, D1062–D1067.